# Use of 3-Dimensional Videography as a Non-Lethal Way to Improve Visual Insect Sampling

**Michael F. Curran** [1,2,3,4,*], **Kyle Summerfield** [5]**, Emma-Jane Alexander** [5]**, Shawn G. Lanning** [6]**, Anna R. Schwyter** [1]**, Melanie L. Torres** [1,2]**, Scott Schell** [1]**, Karen Vaughan** [1,2]**, Timothy J. Robinson** [7] **and Douglas I. Smith** [1]

[1] Department of Ecosystem Science and Management, University of Wyoming, Laramie, WY 82071, USA; aschwyte@uwyo.edu (A.R.S.); mtorres4@uwyo.edu (M.L.T.); sschell@uwyo.edu (S.S.); karen.vaughan@uwyo.edu (K.V.); dsmith59@uwyo.edu (D.I.S.)
[2] Program in Ecology, University of Wyoming, Laramie, WY 82071, USA
[3] Wyoming Reclamation and Restoration Center, University of Wyoming, Laramie, WY 82071, USA
[4] Department of Wildlife, Fisheries and Aquaculture, Mississippi State University, Starkville, MS 39759, USA
[5] Shell 3D Visualization Center, University of Wyoming, Laramie, WY 82071, USA; ksummerf@uwyo.edu (K.S.); emmajane.alexander@uwyo.edu (E.-J.A.)
[6] Wyoming Geographic Information Science Center, University of Wyoming, Laramie, WY 82071, USA; sgl55@uwyo.edu
[7] Department of Mathematics and Statistics, University of Wyoming, Laramie, WY 82071, USA; tjrobin@uwyo.edu
[*] Correspondence: curran.mf@gmail.com

**Abstract:** Insects, the most diverse and abundant animal species on the planet, are critical in providing numerous ecosystem services which are significant to the United Nation's Sustainable Development Goals (UN-SDGs). In addition to the UN-SDGs, the UN has declared the period 2021–2030 as the "Decade on Ecosystem Restoration". Insects, because of the ecosystem services they provide, are critical indicators of restoration success. While the importance of insects in providing ecosystem services and their role in helping fulfil the UN-SDGs is recognized, traditional techniques to monitor insects may result in observer bias, high rates of type-I and type-II statistical error, and, perhaps most alarmingly, are often lethal. Since insects are critical in maintaining global food security, contribute to biological control and are a key food source for higher trophic levels, lethal sampling techniques which may harm insect populations are undesirable. In this study, we propose a method to visually sample insects which involves non-lethal 3-dimensional video cameras and virtual reality headsets. A total of eight observers viewed video captured insects visiting floral resources in a landscaped area on a university campus. While interobserver variability existed among individuals who partook in this study, the findings are similar to previous visual sampling studies. We demonstrate a combination of 3D video and virtual reality technology with a traditional insect count methodology, report monitoring results, and discuss benefits and future directions to improve insect sampling using these technologies. While improving quantitative monitoring techniques to study insects and other forms of life should always be strived for, it is a fitting time to introduce non-lethal sampling techniques as preservation and restoration of biodiversity are essential components of the UN-SDGs and the "Decade on Ecosystem Restoration".

**Keywords:** 360-degree data capture; biodiversity; conservation; ecological restoration; environmental monitoring; pollinators; UN Sustainable Development Goals; virtual reality; virtual reality display

## 1. Introduction

Multiple global environmental threats have recently led the United Nations to declare the period 2021–2030 as the 'Decade on Ecosystem Restoration' [1] (www.decadeonrestoration.org). These threats are wide ranging and include issues associated with climate change, food security, water quality, and decreases in biodiversity [2]. The UN Sustainable Development Goals (UN-SDGs) (www.sustainabledevelopment.un.org) aim to reduce anthropogenic factors which may contribute to these threats. Insects provide basic ecosystem services which may help achieve UN-SDGs [3]. They are the most diverse and abundant animals on Earth and provide multiple ecosystem services such as serving as a source of food for higher trophic levels [4] and humans [5], inspiring art and culture [6], nutrient cycling [7], maintenance of genetic diversity of plants through pollination [8], biological control [9], and pollination services for crops [10]. While maintaining these ecosystem services is important for overall environmental well-being, pollinator decline is perhaps the most widely publicized issue facing insects worldwide [2]. Healthy pollinator populations are crucial in maintaining global food security, as many crops rely on pollination events to produce a harvestable crop [2]. Though less studied, pollinators play key roles in wild plant reproduction [11] and therefore are also critical to sustaining healthy vegetation communities in non-crop ecosystems.

The United Nations' Global Assessment Report on Biodiversity and Ecosystem Services reported that nearly one million plant and animal species are at risk of extinction, the majority of these being insects [1]. While the importance of restoring habitat for insects, especially endangered or threatened species, is well recognized in crop systems, less attention has been focused on the role of ecosystem restoration to increase pollinator habitat in non-crop ecosystems [12,13]. Enhancing pollinator habitat has benefits to overall ecosystem services within a given landscape [14]. Similarly, quantitative studies have shown how restoration efforts can improve pollinator habitat, diversity, and abundance in ecosystems including tropical forests [15], temperate grasslands and prairies [13,16], heathlands [17], and a variety of other non-crop ecosystems including urban and suburban flower gardens [18]. Additionally, research has documented and quantified increases to pollinating insect populations in crop ecosystems [19]; however, less work has been focused in wild ecosystems [11,12].

Insects are an attractive animal group to study to gauge success of restoration projects due to their ecological importance and because large quantities (i.e., statistically valid) of insects can be obtained in a relatively small amount of time [20]. While ecosystem restoration activities are necessary to improve pollinator habitat in both agriculture and wild ecosystems worldwide, current insect quantification methodologies fail to wholly preserve existing habitats and populations. Common methods to quantify insects in field studies include visually counting floral visits by insects, sweep netting, pit-fall trapping, vacuum netting, sticky trapping, pan or cup trapping by bright colored containers with drowning solution, and flight intercept traps such as Malaise traps [21,22]. Visual counts are subject to high levels of observer bias [23], which may reduce ability for land managers and policy makers to rely on those data. Standard visualized sampling methods also require an observer to be present at a location which may attract or repel insects on plants of interest (e.g., observer height and movement, bright clothing or certain scents may attract or repel insects to or from the study area) and result in high type-I or type-II statistical errors. Additionally, human visualization methods are limited to insect movement in the observer's limited range of view (e.g., an individual flower or small patch of flowers) for a fixed time window. Traditional sweep net techniques, aside from also being subject to high type-I or type-II statistical errors, may not be appropriate for quantifying insects with the goal of conserving or restoring their populations as they require insects to be killed for later classification in a laboratory and often result in the destruction of vegetation encountered along the sweep netting path. Sticky traps and pan traps are typically brightly colored, which may attract insects which may not be typical visitors to the area that they are located in and result in high type-I error rates, and both methods are lethal to the insects collected by them and require special authorization in many countries and protected areas. Other issues associated with all methods are their inability to gain a solid understanding of insect movement patterns, floral preference (e.g., it is impossible to determine whether an insect captured

by a sweep net, sticky trap, or bee cup was visiting a specific plant or portion of a plant when it was caught), and all rely on multiple visits to field sites by individuals involved in this study. Aside from these techniques being time consuming (and therefore labor-intensive and expensive), insects have wide-ranging changes to behavior and location at hourly, daily, and seasonal time scales, resulting in a wide variation in collection results based on sample time and phenology differences [24]. In addition to, or as a result of, the aforementioned issues with each technique, highly variable results are obtained with each method [25,26]. Development of a non-lethal (to insects) and non-destructive (to plants), passive, repeatable sample method, other than visual human observers, would likely be beneficial to properly sample endangered or threatened pollinators and other insects.

Here, we use a 3-dimensional (3D), 360-degree camera to perform data capture of a wildflower garden area. This allowed for a one-minute 360-degree video to be captured of the area. The 360-degree video was then experienced by eight observers with a virtual reality (VR) headset. The use of 3D technology has shown to be an effective and non-destructive way to sample a variety of ecosystem services [27,28], but it is seldom used to monitor insects outside of studies to track honeybees at beehive entrances [29]. We sought to examine interobserver variability when the same video was played with eight observers looking into a VR headset at four different angles (rotating 90 degrees from start point to obtain each angle). We also aimed to determine whether differences occurred in insect counts when the video was viewed at different angles of view (i.e., were there differences in insect counts on different flower types?). We discuss our results, highlight advantages of using 3D video cameras over traditional techniques to monitor pollinators and insects for conservation or restoration purposes, and highlight future possibilities of using 3D cameras to examine insect and pollinator populations.

## 2. Materials and Methods

### 2.1. Study Area

Our video collection took place in a wildflower garden outside of the University of Wyoming's Energy Innovation Center (Laramie, WY, USA, 41.3149° N, 105.5666° W). Elevation at the study site is ~2200 m above sea level. This study took place on 16 August 2018 at 14:00 (Mountain Standard Time). This study was conducted with minimal wind speed (<5 kph), in sunshine conditions, at 19 °C. The video observation portion of our study took place inside the Shell 3D Visualization Center, located in the Energy Innovation Center at University of Wyoming, Laramie, WY, USA.

### 2.2. Use of 3D Camera

The camera used in this study was an Insta360 Pro (Shenzhen Arashi Vision Co., Ltd., Guangdong, China). The camera was mounted on a tripod at 1 m above ground level, placed in the center of a diverse floral section of the garden and leveled on the tripod. After camera placement, observers moved outside of the study area and waited ten minutes before recording a video to avoid any insect 'flushing' effects caused by human motion within the garden plot itself. Total recording time spanned approximately ten minutes with different settings, from which one minute of video was selected for this study.

### 2.3. Video Processing

The video was composited from six individual 6k lenses, producing a stereoscopic 8k video. The stitching software is provided by the manufacturer of the Insta360 Pro (Insta360 Pro STITCHER, Shenzhen Arashi Vision Co., Ltd., Shenzhen, China).

### 2.4. Observer Trials with Virtual Reality Headset

A total of eight observers participated in this study. All observers were professors, academic research professionals, or graduate students at University of Wyoming. Four of the observers have extensive (three PhD and one MS level) training in entomology, while the remaining four all have conducted environmental surveys at least at the MS level in other ecological-related fields

(two working to obtain an MS in Ecosystem Science and Management, one with an MS in Geography related to environmental studies, and one working to obtain a PhD in Ecology). The VR headset used for video analysis was the HTC Vive Pro (HTC Corporation, New Taipei, Taiwan). A quiet, temperature-controlled room was available to each observer and video technician during their individual trial period, with others waiting in a separate room. Prior to beginning insect VR counts, each observer received a basic VR headset tutorial and was given a chance to gain comfort using the system by exploring an indoor 3D environment. Each observer received instructions to record each time they saw an insect visit a flower in their field of view using a handheld manual counter. All observers started at a fixed view and watched the video for one minute. Afterwards, the video technician recorded the number of counts, reset the counter, and instructed the observer to turn 90 degrees to their right to obtain the second field of view. The VR headset has a 110° field of view, so some overlap between views was expected when observers rotated 90° between each view. The process was completed a total of four times for each observer to obtain a 360-degree view of the video (Figure 1). Mean counts for each view were estimated using Poisson regression.

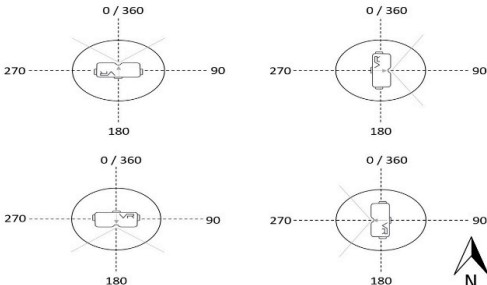

**Figure 1.** Orientation of VR headsets on observers during each one-minute view within the VR display room. Note the field of view from the headset is 110°, so some overlap between views was expected at each view point.

## 3. Results

### 3.1. Variability among Observers

There was high variation among observers (Figure 2). All observers counted the most insects in view 2 and the least insects in view 1, with views 3 and 4 being the second and third highest counts per observer, respectively. Differences in counts among observers could be due to various comfort levels using VR headsets or various experience counting insects. Observer 1 almost always had the highest count per view, whereas observers 3, 6, and 7 were always lower than other observers in insect counts per view.

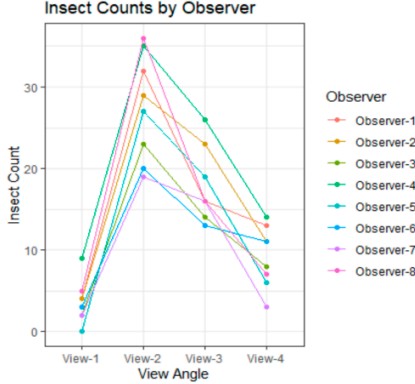

**Figure 2.** Number of insect visits to vegetation binned by each one-minute view for each observer. Note, all observers counted the most insects in view 2, followed by view 3, then view 4, with view 1 always being the lowest count per observer.

## 3.2. Difference among Views

The estimated mean count, standard deviation of counts, and coefficients of relative variation for each of the views are presented in Table 1. Note that all views had significantly different mean counts as evidenced by the non-overlapping confidence intervals on the mean counts in row 1 of Table 1. Although interobserver variability was high, individual observer counts were consistent for views 2–4 when comparing the coefficient of variations (i.e., $\sigma_i/n$, with $\sigma_i$ denoting the standard deviation in counts across observers in group $i = 1, \ldots, 4$ and $n = 8$ observers) ($p = 0.2924$) using the Feltz–Miller test for equivalencies of coefficient of variation across observers [30]. The Feltz–Miller test was done using the *cvequality* package in Program R [31]. Note from Table 1 that the coefficient of variation for view 1 was substantially higher than the coefficients of variation in the other three views. Figure 3 shows the distribution of counts for each of the four views.

**Table 1.** A table depicting results from the Feltz–Miller test. All views had significantly different mean counts as evidenced by the non-overlapping confidence intervals in row 1. Row 2 represents standard deviation and row 3 represents the coefficient of variations.

|  | View 1 | View 2 | View 3 | View 4 |
| --- | --- | --- | --- | --- |
| Mean, 95% C.I. | 3.625, [2.3, 4.94] | 27.625, [23.98, 31.27] | 17.875, [14.95, 20.80] | 9.125, [7.03, 11.22] |
| std. deviation | 2.669 | 6.545 | 4.568 | 3.758 |
| CV | 0.736 | 0.237 | 0.253 | 0.411 |

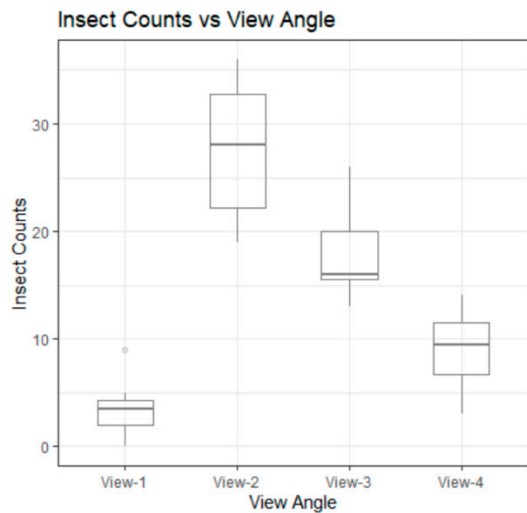

**Figure 3.** Number of insect visits to vegetation binned by each one-minute view for all observers. Note that view 1 had significantly less mean insect counts than all other views.

## 4. Discussion

Our study found statistical support towards using 3D video cameras with 360° view as an alternative tool for quantifying insect behavior compared to traditional techniques. When the video clips were viewed by observers separately, there was variability in counts conducted by different individuals. Our results were similar to previous studies using traditional visual sampling techniques [24]. The larger interobserver variability in view 1 is likely due to the low abundance of insects in this view and the fact that all observers began the experiment with view 1 and there could have been different levels of comfort with the equipment across observers at the beginning of the experiment. However, all observers noticed the most floral hits in the second view (dominated by *Perovskia atriplicifolia* 'Little Spire', Figure 4), the least hits in the first view (dominated by *Potentilla fruticosa* 'Katherine Dykes', Figure 4), and the second and third most hits in views 3 (mixed between *Amelanchier grandiflora* 'Autumn Blossom' and *Perovskia atriplicifola* 'Little Spire') and 4 (mixed between *Potentilla fruticosa*

'Katherine Dykes' and *Perovskia atriplicifolia* 'Little Spire', Figure 4), respectively. The ability to quantify insect activity in a 360° field of view of simultaneous video is not possible with traditional techniques, which makes the use of 3D cameras even more attractive for sampling pollinators and other insects. The 3D view may be especially useful because insect presence and abundance is typically not only related to floral characteristics in a given area, but also to factors such as sunlight, temperature and wind [32].

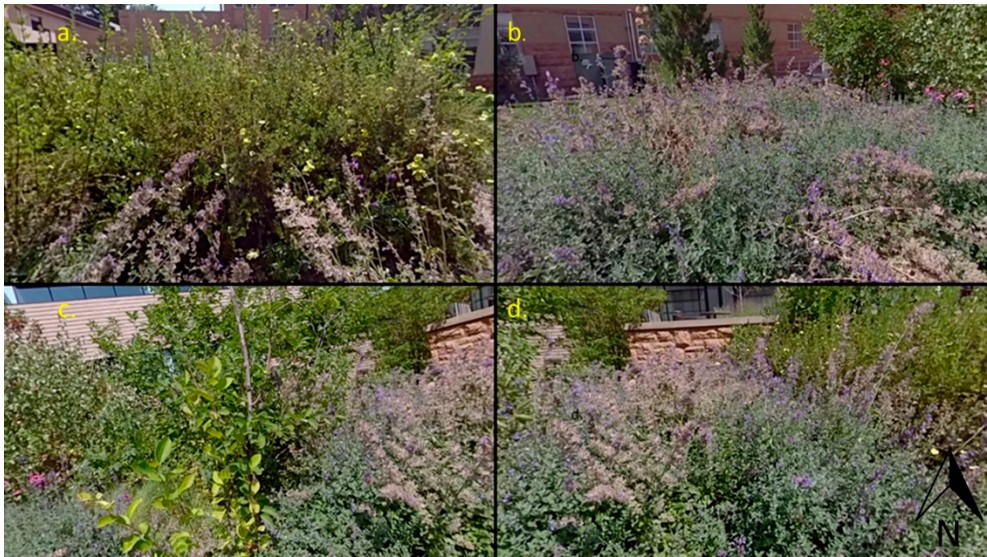

**Figure 4.** Actual view from each angle. View 1 (**a**) was dominated by *Potentilla fruticosa* 'Katherine Dykes'. View 2 (**b**) was dominated by *Perovskia atriplicifolia* 'Little Spire'. View 3 (**c**) was mixed between *Amelanchier grandiflora* 'Katherine Dykes' and *Perovskia atriplicifolia* 'Little Spire'. View 4 (**d**) was mixed between *Perovskia atriplicifolia* 'Little Spire' and *Potentilla fruticosa* 'Katherine Dykes.'.

The camera used in this study should not cause insects to be falsely attracted to or repelled from the sensor as it remains still, is a neutral color (black), is not noticeably taller than surrounding vegetation, and does not have a reflective lens [33]. The camera can also be placed in a fixed area and controlled remotely away from the area of interest while capturing simultaneous video in 3D. This makes the use of 3D cameras an attractive alternative to traditional monitoring techniques, which are usually lethal, disruptive to insects, baiting insects away from their natural resource, limited to one field of view or a combination of the negative aspects associated with traditional insect sampling techniques.

Several caveats exist with this study. First, our study was limited to a single location at a fixed point in time. This study was conducted in near optimal environmental conditions, which likely impacted insect behavior and the overall outcome of this study. However, insect sampling using traditional techniques is also highly subject to environmental conditions (e.g., sweep netting the same location on consecutive days would result in significantly different results if rain or high wind occurred on day one and sunshine and low wind conditions occurred on day two, or if the technician collecting the sweep net sample changed between days). The permanent record of the 3D camera may be beneficial to finely examine environmental conditions at fixed site at multiple times and because the same camera could be placed in the same exact location with the same settings at multiple times, which is near impossible to expect from a human insect collection. Second, VR and insect sampling expertise was highly variable within the observers who participated in this study. Increased training in both areas may have resulted in a reduction in observer bias. However, it should not be assumed that all technicians in large-scale field studies will have the same experience in entomological surveys. Additionally, other studies have shown environmental field surveys are highly subject to observer biases when different technicians are subject to different environmental factors (e.g., cold, rain, and wind) or personal fatigue [34]. The ability

to examine video in a quiet, temperature-controlled environment is another factor making 3D cameras and VR technology appealing for insect sampling.

Similar to visual field sampling, another limitation of our study was that observers were directed to count floral visits by insects rather than to identify individual insects to a genera- or species-specific level. The resolution of the camera used in this study would, however, allow for most insects to be identified beyond the 'insect' level and provides a means to permanently store the visual records without damage to any live insects. Future studies to test this technology will focus on playing video at different speeds or allow multiple observers to view the same video angle simultaneously; these changes may result in reduced observer bias or the ability to identify insects to finer taxonomic levels compared to traditional field count techniques. Also promising are artificial intelligence and machine learning, which may allow for the computer software to count and identify insects (similar to facial recognition algorithms used to detect humans on security cameras). However, this would require sufficient training of the computer, which would be time consuming, costly and would likely require an ample amount of collected insects for algorithms to become accurate, since insect identification to the species-specific level often relies on examining very fine morphological characteristics.

The ability to interact with data in a controlled environment with multiple individuals has also shown to be a useful educational tool [35], which is another advantage of having a permanent video record available for viewing by multiple observers in a controlled setting. In this instance, the group of observers could potentially look at the video together and determine errors (e.g., why observer 1 always seemed to have higher counts than others and observers 3, 6 and 7 seemed to have lower counts than others). Other studies have shown that the use of cameras recording individuals collecting insects have been useful to correct observer bias [36] and that images provide permanent records which can be reviewed by multiple observers and stakeholder groups in vegetation surveys [37]. While our study was limited to insects, the technique presented could also be used as a non-lethal sampling strategy to study behavior in other animals (e.g., monitoring migratory bird behavior at stopover grounds). The camera used in this study performs optimally during daylight hours, but as technology continues to develop, it is likely 3D video cameras with sensors to capture data in low-light situations could be useful for monitoring nighttime behavior of insects and other animals.

## 5. Conclusions and Future Research

Biodiversity and pollinator declines will continue to be global environmental issues for the foreseeable future [2]. Therefore, implementing non-lethal monitoring techniques will be critical towards improving our understanding of insects without harming their populations. Additionally, using new quantification tools in restoration ecology can provide further insight towards selecting specific seed mixes and vegetation communities that promote pollinator presence and use [38] and may help promote seed mixes which attract insects desirable to bird and other wildlife species as forage [39]. This technology may also aid in future monitoring of endangered insects. For example, the United States Fish and Wildlife Service restricts destructive insect sampling techniques within core habitat areas of *Bombus affinis* (Rusty Patched Bumble Bee) [40]. While the need to select seed mixes and vegetation communities which promote pollinator diversity and insect populations in restoration projects has been documented [39], techniques to quantify insect activity on specific vegetation and floral structures should help improve understanding of how pollinators are utilizing areas designated for restoration and will aid land managers and restorationists with future decision-making.

Although our study was limited to a single spatio-temporal location, we have demonstrated the ability of a 3D camera and VR technology to provide an adequate way to sample pollinator floral hit counts in a non-lethal or invasive way. We have demonstrated examples of how 3D cameras may provide benefits compared to other traditional sampling techniques. The major benefits we have focused on are the ability of a camera to capture a 360-degree field of view to obtain a non-lethal sample of insects in a given space over a given time, the ability of a camera to provide permanent video records which can be observed or re-analyzed by multiple individuals, and the ability of a camera to remain

still with neutral color to reduce false attraction or repulsion of insects in a study area. As technologies rapidly advance, there will surely be more utility for 3D cameras and VR technology for future studies. Currently, the cost of this technology may be limiting, though VR technology is becoming more widely accessible to a wide range of potential users. Future research to study the economics of this technology will be useful, and as technology continues to develop, we expect this to become more widely available and less cost prohibitive.

Exciting areas of future research will be to utilize machine learning algorithms to track insect movement and behavior, an area which is recognized as important for insect conservation and overall knowledge of insects [41,42]. The resolution of the camera in this studied coupled with machine-learning algorithms has potential to allow for individual insects within each video clip to be examined based on their behavior and choices of floral selection. It may also allow for commonalities and differences of behavior to be better understood within and among different insect guilds. Many additional interaction technologies such as immersive 3D Cave environments, augmented reality displays should be explored as methods to assist in non-invasive insect counts and as ways to train new technicians before they are tasked with insect count projects. Other studies have shown that the use of 2D video to track technicians sampling in the field have been useful in correcting common collection mistakes [36]. The 3D cave environment would likely prove useful as an educational tool since it can be used similarly to a classroom setting, allowing multiple individuals to be involved in interactive training. The VR headsets in this study limit the ability of users to see the view of other users, though a 3D cave environment may function more like a classroom setting by allowing all users to see the same view simultaneously [43].

While our study was limited to a fixed time, with the goal of analyzing the 3D video with multiple observers, keeping the camera in a fixed place with a longer recording period or with a programmed setting to record for short durations at different intervals throughout a longer time period (e.g., a day, a week, a growing season) would allow for better understanding of insect behavior and activity over time. This could be critical for future research as one of the most difficult aspects of insect sampling (compared to vegetation and larger wildlife) is their almost constant spatial transitioning over time [25]. Additionally, spacing more than one camera out over time and space could be very influential to future restoration practices as understanding how insect groups utilize different floral resources at different phenological periods is a critical area of pollinator habitat research in restoration ecology [36]. Finally, since insects play such important roles in various ecosystem services and because recent research suggests declines in insects, especially pollinators, lead to limitations in crop production [44], developing non-lethal sampling strategies to better understand insects will be beneficial to helping document without destroying these critical ecosystem engineers will likely be favorable to the UN-SDGs.

**Author Contributions:** The study concept was developed by M.F.C.; E.-J.A. and K.S. provided technological expertise and access to facilities. All authors aside from E.-J.A., K.S., and T.J.R. participated in this study as visual observers. M.F.C. wrote first draft, and all authors contributed edits and participated equally in further drafts. Specifically, D.I.S., K.V. and S.S. provided entomological expertise. E.-J.A., K.S., S.G.L. and K.V. provided advice about virtual reality. T.J.R. provided advice on statistical writing and statistical design. M.L.T. and A.R.S. provided general ecological expertise and grammatical advice. Figure 1 was made by S.G.L., Figures 2 and 3 were made by T.J.R. and A.R.S., and Figure 4 was made by M.F.C. All authors have read and agreed to the published version of the manuscript.

**Funding:** This research received no external funding.

**Acknowledgments:** We thank the School of Energy Resources and the Shell 3D Visualization Center at University of Wyoming for providing technology to conduct this study. We also thank the Wyoming Reclamation and Restoration Center at University of Wyoming for providing resources to support research relating to ecological restoration. We are grateful for Ruben Aleman for participating in this study as an observer.

**Conflicts of Interest:** The authors declare no conflict of interest.

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
