# Peer review of "Use of 3-Dimensional Videography as a Non-Lethal Way to Improve Visual Insect Sampling"

_land, doi:10.3390/land9100340_

Round 1

Reviewer 1 Report

Dear Editor and Authors

First of all, thanks for the opportunity to read and evaluate this manuscript. The authors use a technology that is becoming accessible to more people to develop a new form of insect sampling. The main advantages of the methodology are non-lethality and less interference in insects' behavior about conventional methods. The article is interesting; however, it has limitations pointed out by the authors themselves, which I agree with them.
Below, it is presented my doubts and suggestions to improve the manuscript. The intention is not to discourage them, but to raise doubts and encourage the improvement of the presented technique.

Abstract: It is not easy to follow the link between (i) the United Nation's Sustainable Development Goals (UN-SDGs), ecosystem services, and insect sampling techniques. The authors should be more objective and directly point out the main problem presented in the article, that is, the need to improve insect sampling techniques.
Material and Methods
L112 - The study was performed in optimal conditions; minimal windspeed (<5 kph), in sunshine conditions, at 19 ° C. However, these conditions are hardly found in many situations and places where sampling needs to be done. As one of the scopes of the journal encourages research ideas, the article fits this profile perfectly. However, while I find the development of this sampling technique exciting and reasonably promising, it is necessary to consider its various limitations. Only in some situations could this technique, with the available resources, be applied in larger-scale studies.

Results
L148 - The authors said: "There was high variation among observers." In the Material and Methods sections, no information on the degree of insect knowledge of each observer was presented. At a university, we can find people able to identify species from various insect families, while others may be unable to differentiate insect orders. Likewise, people with greater interest and intimacy with entomology should more easily perceive the presence of insects. Thus, more information about the eight observers is needed.

In this experiment, the objective was to detect floral visitors without determining their specific level. However, for this particular purpose, the 3-D camera can be overstated, since, with other equipment, it would be possible to perform the same task.
L158-171 - Statistical analysis should be described previously in the Material and Methods section;

Discussion
L178-180 - The authors begin the discussion by emphasizing the importance of the results for the study of insect behavior. While the Abstract and Introduction address the problem of insect sampling in general, a more precise purpose is presented in the discussion in which the 3-D videography technique would be suitable. The authors should limit the purpose for which the technique would be most appropriate. For example, the technique may not be the most appropriate for sampling to estimate insect pests' economic threshold level or for large-scale ecological studies of insect diversity.

Author Response

Please see attachment.  Our comments are highlighted in yellow and our changes to the revised manuscript were also highlighted in yellow.  Thank you for you feedback, we feel your comments improved our revised version.

Reviewer 2 Report

The use of some terminology need to be reviewed. Please check the suggestions done in the manuscript.

The text lack of criticism about the limitations of the technique, in special the difficulty of identifying many species based only on pictures. Weakness should be also included in the discussion.

A reference to the North is needed in all the pictures; there is no mention about the orientation-effect in insects presence and activity

Many references need to be adequated to the journal instructions been they marked or not

Author Response

Please see attached.  We replied to your comments in text highlighted in yellow, our revised manuscript also has highlights where we made edits.  We appreciate your help in improving this manuscript.

Reviewer 3 Report

An obviously appropriate methodology for fixed-site monitoring, as in gardens, with potential applications for use in transect studies and the like. The attention paid to interobserver variance was much appreciated.

Author Response

Thank you for your review and your praise.  We have addressed issues brought up from other reviewers.

Reviewer 4 Report

This study describes an exploratory application of 3D 360-degree video to sample pollinator activity. While it is based on a single brief video and a handful of naïve observers, the results suggest that this technology could be useful in certain situations. However, there are considerable limitations on use of this technology, and the manuscript should include an explicit description of this, such as cost of equipment and data management, technology support requirements, and power supply. Additionally, this technique is subject to the same severe limitation of taxonomic identification resource bandwidth facing other insect sampling approaches. Still, for applications with potential access to funding, this does represent a promising possibility. I recommend including a description of the potential for deployment in citizen science applications, assuming future widespread availability of VR headsets. Finally, this is of course not limited to use with insects, or pollinators. I imagine it would be very useful (provided funding of course) for studying any taxa in situations involving a specific location: documenting visits to watering holes, monitoring migratory stopovers, observing behavior of juveniles at dens, etc. Use of cameras sensitive in very low light would be particularly useful.

Minor comments:

Please include a brief description of the Feltz-Miller test and an explanation of why it is appropriate for this study.

Table 1 should have a descriptive caption.

Line 258: provide a description of CAVE environments.

Author Response

Thank you for your review.  We have attached a file with our comments highlighted in yellow.  Our revised manuscript also has changes highlighted in yellow for reviewers/editor to be able to easily track our revision.

Round 2

Reviewer 1 Report

Dear Editor and Authors,
I have read the corrected version of the manuscript. The authors answered my questions and included my suggestions. Therefore, I recommend publishing the article in the present form.
Sincerely,